# Incidence of *Carassius auratus* Gibelio Gill Hemorrhagic Disease Caused by CyHV-2 Infection Can Be Reduced by Vaccination with Polyhedra Incorporating Antigens

**DOI:** 10.3390/vaccines9040397

**Published:** 2021-04-16

**Authors:** Tingting Zhang, Yuchao Gu, Xiaohan Liu, Rui Yuan, Yang Zhou, Yaping Dai, Ping Fang, Yongjie Feng, Guangli Cao, Hui Chen, Renyu Xue, Xiaolong Hu, Chengliang Gong

**Affiliations:** 1School of Biology and Basic Medical Sciences, Soochow University, Suzhou 215123, China; zhangtt199403@163.com (T.Z.); 20185221003@stu.suda.edu.cn (Y.G.); 20174221043@stu.suda.edu.cn (Y.D.); yjfeng@suda.edu.cn (Y.F.); caoguangli@suda.edu.cn (G.C.); xuery@suda.edu.cn (R.X.); 2Jiangsu Center for Control and Prevention of Aquatic Animal Infectious Disease, Nanjing 210036, China; xiaohanliu1975@163.com (X.L.); yr8624@163.com (R.Y.); jsscykzx@163.com (P.F.); chenhuijsbf@163.com (H.C.); 3Dafeng District Aquaculture Technical Extension Station of Yancheng City, Yancheng 224100, China; 99142028@163.com; 4Agricultural Biotechnology Research Institute, Agricultural Biotechnology and Ecological Research Institute, Soochow University, Suzhou 215123, China

**Keywords:** CyHV-2, vaccine, polyhedral microcrystals, cypoviruspolyhedra

## Abstract

Encapsulation of antigens within protein microcrystals (polyhedra) is a promising approach for the stable delivery of vaccines. In this study, a vaccine was encapsulated into polyhedra against cyprinid herpesvirus II (CyHV-2). CyHV-2 typically infects gibel carp, *Carassius auratus* gibelio, causing gill hemorrhagic disease. The vaccine was constructed using a codon-optimized sequence, D4ORF, comprising the ORF72 (region 1–186 nt), ORF66 (region 993–1197 nt), ORF81 (region 603–783 nt), and ORF82 (region 85–186 nt) genes of CyHV-2. The H1-D4ORF and D4ORF-VP3 sequences were, respectively, obtained by fusing the H1-helix sequence (region 1–90 nt) ofBombyx mori cypovirus(BmCPV) polyhedrin to the 5′ terminal end of D4ORF and by fusing a partial sequence (1–279 nt) of the BmCPV VP3 gene to the 3′ terminal end of D4ORF. Furthermore, BmNPV-H1-D4ORF-polh and BmNPV-D4ORF-VP3-polh recombinant *B. mori* nucleopolyhedroviruses (BmNPVs), belonging to the family Baculoviridae, and co-expressing BmCPV polyhedrin and H1-D4ORF or D4ORF-VP3, were constructed. H1-D4ORF and D4ORF-VP3 fusion proteins were confirmed to be encapsulated into recombinant cytoplasmic polyhedra by Western blotting. Degradation of vaccine proteins was assessed by SDS-PAGE, and the results showed that the encapsulated vaccine proteins in polyhedra could be protected from degradation. Furthermore, when gibel carp were vaccinated with the purified polyhedra from BmNPV-H1-D4ORF-polh and BmNPV-D4ORF-VP3-polh via injection, the antibody titers in the serum of the vaccinated fish reached 1:6400–1:12,800 at 3 weeks post-vaccination. Therelative percentage of survival of immunized gibel carp reached 64.71% and 58.82%, respectively, following challenge with CyHV-2. These results suggest that incorporating vaccine protein into BmCPV polyhedra may be a novel approach for developing aquaculture microencapsulated vaccines.

## 1. Introduction

Aquaculture vaccines include live, inactivated, and genetically engineered vaccines. Of the three, genetically engineered vaccines can be further classified into recombinant subunit vaccines, DNA vaccines, gene deletion/mutant vaccines, and living vector vaccines according to the preparation method [1]. These vaccines have played an important role in reducing the incidence of aquatic diseases. However, the safety and the vaccination route and delivery methods of vaccines in aquaculture need further improvement. Occasionally, under natural and aquaculture conditions, reversion to virulence of the live attenuated vaccines leads to reduced safety [2]. Inactivated vaccines usually do not provide immune protection that is as effective as live vaccines; therefore, appropriate adjuvant and repeated vaccinations are often required [1,3]. Subunit vaccines are considered to be degraded rapidly in vivo, thus booster shots are required to get ongoing protection against diseases [3,4]. DNA vaccines are poorly transported into cells, therefore, physical methods (electroporation, needle-mediated injections, intranasal administration, and gene gun) and nonviral methods (lipid- and polymer-based methods) are used to deliver DNA vaccines into target cells. In addition, due to exposure to various enzymes (i.e., nucleases) and to reactive oxygen species, DNA vaccines are easily degraded in vivo, which may decrease their bioavailability [5]. Moreover, vaccines based on DNA plasmids often contain antibiotic resistance genes, and the potential integration of such a sequence into the host genome may represent a risk [6].

In recent years, novel vaccine formulations have been developed by using controlled release technology to improve the safety and effectiveness of vaccine delivery systems (VDSs). Toward that end, biodegradable sustained-release microcapsules are being given much attention internationally in VDS research. Nanoparticles such as nanospheres, nanocapsules, and nanomicelles, as modes of VDSs, have been generated through the combination of nanotechnology with polymer materials and polymer chemistry technology. Nanoparticle VDSs have previously been used in aquaculture [7,8,9,10,11]. Among various nanodelivery systems, chitosan, polylactic-co-glycolic acid (PLGA), and sodium alginate (S-Alg), as synthetic and natural polymers, are the most widely used carriers for controlled release due to their biocompatibility and biodegradable properties. These products have also shown low toxicity and are available at a relatively low cost [11,12,13].

Polyhedrons produced by Bombyx mori cytoplasmic polyhedrosis virus (BmCPV) are protein microcrystals, which may serve as a candidate VDS. The BmCPV genome consists of ten discrete double-stranded RNA segments (S1–S10) [14]. S1, S2, S3, S4, S6, and S7 segments of BmCPV encode viral structural proteins. S5, S8, S9, and S10 segments encode non-structural proteins p101 (NSP5), p44 (NSP8), NS5 (NSP9), and polyhedrin, respectively [14]. Protein microcrystals (occlusion bodies), termed polyhedra (about 2–5 µm in diameter), containing several thousand virus particles, can be produced in the cytoplasm of infected cells. The polyhedra have a protective effect on the embedded virus particles, providing resistance to harsh environmental conditions [15]. Previous studies have shown that the addition of a tag of 75 amino acid residues (VP3 tag), derived from the BmCPV turret protein (TP) coded by the S4 segment, to the N-terminus of foreign proteins allowed for their incorporation into polyhedra. Likewise, a smaller 52 amino acid segment spanning residues 42–93 of the TP has been reported to be the minimum tag required for immobilization of recombinant proteins into polyhedra [16]. The atomic structure of BmCPVpolyhedra showed that the polyhedrin H1-helix (the H1 tag, with 30 amino acid residues at the N-terminus) of the molecule is a useful tag for incorporating recombinant proteins into polyhedra such as the VP3 immobilization signal [17]. Thus far, the major capsid protein VP1 of norovirus-like particles [18], vascular endothelial growth factor [19], endostatin [19], protein kinase C [20], fibroblast growth factors [21,22], leukemia inhibitory factor [23], and bone morphogenetic protein-2 [24] have beenincorporated into polyhedra by co-expression of BmCPVpolyhedrin and the target gene fused with the H1 tag or the VP3 tag. However, until now, there have been no reports on the incorporation of recombinant vaccines into BmCPVpolyhedra or an evaluation of the immune-protective effects of apolyhedron-incorporating vaccine.

The gibel carp, C. auratus gibelio, is one of the most important freshwater fish species in China. An outbreak of C. auratus gibelio gill hemorrhagic disease caused by cyprinid herpesvirus 2 (CyHV-2) infection first occurred in Yancheng City of Jiangsu Province, China, in 2012 [25,26]. At present, this hemorrhagic disease is spreading to other provinces of China because an effective prevention and control system has not yet been established. In the aquaculture industry, immunization is widely considered to be the most effective approach for preventing viral infection in fish. Several vaccines have been reported to protect fish from CyHV-2 infection, including an inactivated vaccine [27,28], recombinant subunit vaccine expressed in Pichia pastoris [29], genetic engineering subunit vaccine based on the baculovirus surface display system [30], and recombinant baculovirus vector vaccine [31].

In this study, an anti-CyHV-2 multi-subunit vaccine protein was incorporated into polyhedra by co-expressing BmCPV polyhedrinand the target gene fused with the H1 tag or the VP3 tag. It was found that the vaccine protein encapsulated in polyhedra can be protected from degradation. In addition, the relative percent survival (RPS) of gibel carp immunized with the polyhedra via an injection route reached 64.71% and 58.82%, respectively, following challenge with CyHV-2. These results suggest that incorporating vaccine protein into BmCPVpolyhedra may be a novel approach for developing aquaculture microencapsulated vaccines.

## 2. Materials and Methods

### 2.1. Preparation of CyHV-2

Kidney and spleen tissues (5 g) dissected from five diseased fish infected with CyHV-2 were homogenized and diluted at a ratio of 1:10 (*w*/*v*) in 1× phosphate-buffered saline (PBS, pH = 7.4). After centrifugation at 8000× *g* for 15min, the supernatant was filtrated through a filter with a pore size of 0.45 μm (Sangon, Shanghai, China). Viral genomic copies of the obtained filtrate were determined by absolute quantitative PCR using the reference plasmid, pMD-hel, containing the partial sequence of the CyHV-2 helicase gene [32]. The viral genome copies of the filtrate were then adjusted to 10^5^ copies/μL to be used as virus stock.

### 2.2. Synthesis of H1-D4ORF and D4ORF-VP3 Fused DNA Sequences

In our previous study, a codon-optimized open reading frame (ORF) sequence, D4ORF, comprising the ORF72 (region 1–186 nucleotides (nt)), ORF66 (region 993–1197 nt), ORF81 (region 603–783 nt), and ORF82 (region 85–186 nt) genes of CyHV-2 was synthesized [31]. In this study, H1-D4ORF and D4ORF-VP3 fused sequences (Appendix A) were obtained by fusing the H1-helix sequence (H1-tag) (region 1–90 nt) [17] of BmCPV polyhedrin (polh) to the 5′ terminal end of D4ORF and by fusing a partial sequence (1–279 nt, VP3tag) [16] of the BmCPV VP3 gene to the 3′ terminal end of D4ORF.

### 2.3. Construction of the Baculovirus Transfer Vector and Recombinant Baculovirus

The ORF of the BmCPVpolh gene was excised from pIZT-V5/His-polh [33] with EcoRI/XbaIandcloned into pFastBacTM Dual (Invitrogen, Frederick, MD, USA) to generate pFastBacTMDual-polh. The H1-D4ORF and D4ORF-VP3 sequences were cloned into the XhoI/KpnI sites of pFastBacTMDual-polh to generate pFast-H1-D4ORF-polh and pFast-D4ORF-VP3-polh baculovirus transfer vectors, respectively.

pFast-H1-D4ORF-polh and pFast-D4ORF-VP3-polh were transformed into Escherichia coli DH10Bac/BmNPV [34] to generate Bacmid-H1-D4ORF-polh and Bacmid-D4ORF-VP3-polh recombinant bacmids, respectively, using the Bac-To-Bac baculovirus expression system (Invitrogen, Frederick, MD, USA) following the manufacturer’s instructions. pFastBacTMDual was used to generate a control bacmid.

Two micrograms of Bacmid-H1-D4ORF-polh or Bacmid-D4ORF-VP3-polh DNA weremixed with 10 µL of FuGENE HD Transfection Reagent (Roche Diagnostics, Indianapolis, IN, USA), and then transfected into a cultured silkworm ovary-derived BmN cell line (105 cells in 200 µL TC-100 medium (Sangon, Shanghai, China) without fetal bovine serum). The transfected cells were cultured with TC-100 medium containing 10% fetal bovine serum at 27 °C after removing the old medium 4 h post-transfection. After 3 days, the supernatant of the cultured cells was collected to obtain recombinant baculoviruses BmNPV-H1-D4ORF-polh or BmNPV-D4ORF-VP3-polh. To further verify the expressed protein by recombinant viruses, 50 μg of total proteins extracted from infected cells were subjected to SDS-PAGE. Proteins were transferred onto polyvinylidene difluoride (PVDF) membranes (Millipore, Burlington, MA, USA), and Western blotting was conducted with a mouse anti-ORF72 (1:500) [31] and an HRP-conjugated goat anti-mouse IgG (1:20,000) (Sino Biological Inc., Beijing, China). Untreated cells were used as a negative control.

### 2.4. Expression of H1-D4ORF and D4ORF-VP3 in the Silkworm 

The newly molted larvae of the fifth instar silkworm were inoculated with BmNPV-H1-D4ORF-polh and BmNPV-D4ORF-VP3-polh by subcutaneous injection, followed by feeding with fresh mulberry leaves at 25 °C. The hemolymph was collected at 1, 2, 4, and 6 days post-injection. Ten microliters of hemolymph were subjected to SDS-PAGE and Western blotting. The hemolymph of healthy silkworm larvae was used as a negative control.

### 2.5. Purification of Polyhedra (Polyhedral Microcrystals)

Forty milliliters of hemolymph collected from 100 infected silkworms at 6 days post-inoculation were used for purification of polyhedra according to our previous report [35]. Briefly, after the hemolymph was centrifuged for 10 min at 6000× *g*, the obtained pellet was washed 3–5 times with 0.1% sodium dodecyl sulfate (SDS) (Sangon, Shanghai, China) until the pellet turned white. The pellet was washed with 1× PBS twice to obtain the purified polyhedra.

### 2.6. Detection of H1-D4ORF and D4ORF-VP3 Recombinant Proteins in Polyhedra and Hemolymph

The hemolymph of the silkworm larvae wascollected 6 days post-infection with recombinant viruses, the supernatant and pellet (polyhedra) were obtained by centrifugation at 6000× *g* for 5 min. The obtained pellet was washed 3–5 times with 0.1% SDS and 2 times with 1× PBS to obtain the purified polyhedra. To dissolve polyhedra, the purified polyhedra were incubated with a lysis buffer (0.2 mol/L Na_2_CO_3_-NaHCO_3_) for 30 min at 30 °C until the suspension became clear and this was followed by adjusting the pH value to 8.0–9.0 with 1 mol/L HCl. Protein (50 μg) from the hemolymph, supernatant, and polyhedrawassubjected to SDS-PAGE [35]. Proteins were transferred onto PVDF membranes and Western blotting was carried out with mouse anti-ORF72 (1:500) and HRP-conjugated goat anti-mouse IgG (1:20,000) (Sino Biological Inc., Peking, China).

### 2.7. Storage Stability of the Recombinant Protein Incorporated into the Polyhedral Microcrystals

The purified polyhedra were divided into four groups and lysed according to our previous paper [35]. In group LPS-20, the lysed polyhedra were stored at −20 °C; in group LPS-RT, the lysed polyhedra were stored at room temperature (about 22 °C); in group PS-20, polyhedra without lysis were stored at −20 °C; and in group PS-RT, polyhedra without lysis were stored at room temperature. After two weeks, 15 μg of the treated samples were subjected to SDS-PAGE. Protein degradation was estimated by electropherogram after the PAGE gel was stained with Coomassie brilliant blue R250 (Sangon, Shanghai, China). 

### 2.8. Vaccination

Gibel carp (mean body weight: 22 ± 2 g) were purchased from Aquatic Breeding Field of Kunshan City, in Jiangsu Province, China. The fish were acclimatized at 25 °C in the laboratory for two weeks in a recirculating freshwater system. Fish were fed with commercial dry pellets (Tongwei Group, Chengdu, China) three times a day.

Sixty fish were randomly divided into three groups (definedas group H1-D4ORF, group D4ORF-VP3, and group CK), with 20 fish in each group. The fish were immunized by injection. In group H1-D4ORF, the fish were vaccinated by intraperitoneal injection with 200 μL of BmNPV-H1-D4ORF-polh polyhedra (7.0 × 10^6^ polyhedra determined by a hemacytometer mixed with complete Freund’s adjuvant at a ratio of 1:1) per fish; in group D4ORF-VP3, the fish were vaccinated by intraperitoneal injection with 200 μL of BmNPV-D4ORF-VP3-polh polyhedra (4.4 × 10^6^ polyhedra mixed with complete Freund’s adjuvant at a ratio of 1:1) per fish; in group CK, the fish were injected with 200 μL of PBS mixed with complete Freund’s adjuvant at a ratio of 1:1 per fish. This final group served as an untreated control.

### 2.9. Enzyme-Linked Immunosorbent Assay (ELISA)

To determine the antibody titers in the immunized fish, blood (*n* = 3) harvested from the caudal vein of fish 3 weeks post-vaccination wasstored overnight at 4 °C. After centrifugation for 10 min at 2000× *g*, the sera were collected and antibody titers were determined according to our previous report [31]. Briefly, ELISA plates were coated with different concentrations of recombinant ORF72, followed by blocking with 5% bovine serum albumin overnight at 4 °C. Plates were incubated with different dilutions of sera from vaccinated or unvaccinated fish for 2 h. After washing five times with Tris-buffered saline containing Tween-20 (Sangon, Shanghai, China), the ORF72 antibody (1:500) was added and incubated for 2 h. After further washing with TBST, HRP-conjugated goat anti-mouse IgG (Sino Biological Inc., Peking, China) was added and incubated. Finally, the absorbance at 492 nm was measured after coloration.

### 2.10. Challenge Trials

Fish 31 days post-vaccination were challenged by injection at the base of the pectoral fin with CyHV-2 stock (10^7^ copies of virus per fish). The challenged fish were raised at 23–25 °C. Five days later, the diseased fish were detected by reverse transcriptionPCR, with the designed primer pairs CyHV-2-hel-1(5′-GGGTGAGGACTTGCGAAGAG-3′) and CyHV-2-hel-2 (5′-CGCTCGTCCGGGTTCTGCACG-3′) based on the helicase gene sequence of CyHV-2 (GenBank accession No. KT387800) to confirm CyHV-2 infection.

The protection effect was evaluated using RPS, calculated by the following formula:RPS (%) = (X − Y)/X × 100(1)
in which X isthe infection rate in group CK, and Y isthe infection rate in the vaccination group. The percentage of infected fish in the test fish group was defined as the infection rate in this study.

### 2.11. Detection of the Incorporated Baculovirus BmNPV into BmCPV Polyhedra

The isolated recombinant polyhedra from silkworm larvae infected with BmNPV-H1-D4ORF-polh or BmNPV-D4ORF-VP3-polh were washed 5 times with 0.1% SDS. The purified polyhedra (10^8^ in 0.5 mL) were lysed with 0.2 mol/L Na_2_CO_3_-NaHCO_3_ buffer (0.5 mL) at 30 °C for 30 min, and mixed with 0.1% SDS (1 mL) or PBS (1 mL), followed by incubation for 30 min at 26 °C. After filtering with a pore size of 0.22 μm, 20 µL of the filtrate were used to infect silkworm BmN cells (2 × 10^5^). After 4 days, incorporation of the recombinant baculoviruses into BmCPV polyhedra was assessed according to morphological changes of the cells by microscopy (Nikon, Tokyo, Japan).

## 3. Results

### 3.1. Generation of Recombinant Baculovirus and Expression of Recombinant Proteins

In order to incorporate recombinant antigen proteins into BmCPVpolyhedra, the baculovirus transfer vectors pFast-H1-D4ORF-polh and pFast-D4ORF-VP3-polh were constructed. In pFast-H1-D4ORF-polh, the H1-D4ORF fused sequence was controlled by the baculovirus P10 promoter, and the BmCPV polh gene was driven by the baculovirus polh promoter. pFast-D4ORF-VP3-polh was generated by replacing the H1-D4ORF fused sequence of pFast-H1-D4ORF-polh with the D4ORF-VP3 fused sequence. Therefore, when the D4ORF with either the H1 tag or VP3 tag is co-expressed with the BmCPV polh gene, the recombinant D4ORF with either the H1 tag or VP3 tag can be encapsulated into polyhedral microcrystals (Figure 1).

To generate a recombinant bacmid, the pFast-H1-D4ORF-polh and pFast-D4ORF-VP3-polh were, respectively, transformed into *E. coli* DH10Bac/BmNPV. The generated recombinant bacmids were termed Bacmid-H1-D4ORF-polh and Bacmid-D4ORF-VP3-polh, respectively. Further, the recombinant baculovirus BmNPV-H1-D4ORF-polh and BmNPV-D4ORF-VP3-polh were generated by transfecting the recombinant bacmid DNAs into cultured BmN cells. The transfected cells ceased growing, but their diameter and nucleus size typically increased and exhibited detachment and lysis 3 days post-transfection. Moreover, the polyhedra could be observed in transfected cells with Bacmid-H1-D4ORF-polh or Bacmid-D4ORF-VP3-polh DNAs (Figure 2A–C). These results indicated that the BmCPV polh gene was expressed and cytoplasmic polyhedra were formed in the transfected cells.

The cultured BmN cells were infected with either BmNPV-H1-D4ORF-polh or BmNPV-D4ORF-VP3-polh, and the collected cell-cultured supernatants were used to inoculate silkworm larvae. The blood color of the silkworm larvae 5 days post-infection became milky white, and polyhedra could be found in their hemolymph (Figure 2D–F). The collected hemolymph 1, 2, 4, and 6 days post-infection was subjected to SDS-PAGE and Western blotting analysis was conducted using mouse anti-ORF72 antibodies. The specific signal bands representing H1-D4ORF (27.94 kDa) and D4ORF-VP3 (34.87 kDa) recombinant proteins could be detected in the hemolymph of BmNPV-H1-D4ORF-polh- and BmNPV-D4ORF-VP3-polh-infected silkworms, respectively (Figure 2G,H).

### 3.2. H1-D4ORF and D4ORF-VP3 Recombinant Proteins Can Be Incorporated into Polyhedral Microcrystals

The results showed that 500 mg of purified polyhedra (wet weight) were obtained from 40 mL of hemolymph collected from 100 silkworms infected with recombinant BmNPV. In order to verify whether H1-D4ORF and D4ORF-VP3 recombinant proteins co-expressed with BmCPV polh were incorporated into polyhedral microcrystals, the collected hemolymph from the silkworm larvae 6 days post-infection were centrifuged and the resultant purified polyhedra (Figure 3A,B), supernatant, and the collected hemolymph were subjected to SDS-PAGE (Figure 3C) and subsequent Western blotting analysis. Specific bands representing target proteins could be observed in the polyhedra samples, indicating that the expressed H1-D4ORF and D4ORF-VP3 recombinant proteins in the hemolymph were incorporated into polyhedra (Appendix A). However, some of the recombinant proteins were free, and located outside the polyhedra, because specific bands representing target proteins could be found in the resultant supernatant (Figure 3D).

### 3.3. Polyhedra Protects the Incorporated Proteins from Degradation

In order to assess the storage stability of the recombinant protein incorporated into the polyhedral microcrystals, the treated polyhedra were subjected to SDS-PAGE. Regardless of whether the polyhedra were lysed or not, there was no significant change in the electrophoretic pattern between the samples (group LPS-20 and group LPS-RT) stored for 2 weeks at −20 °C. However, when the samples (group PS-20 and group PS-RT) were stored at room temperature for 2 weeks, the protein in the lysed polyhedra samples was significantly degraded. There was no significant difference in the electrophoretic patterns of the polyhedra stored at −20 °C (group LPS-20) and the polyhedra stored at room temperature (group PS-RT) (Figure 4). These results suggest that polyhedra protect the incorporated proteins from degradation.

### 3.4. Injection Immunization with Polyhedron-Encapsulated H1-D4ORF/D4ORF-VP3 Fused Proteins Confers Protection against CyHV-2

To explore whether specific antibody (IgM) could be generated in the vaccinated fish by injection of polyhedron-encapsulated H1-D4ORF/D4ORF-VP3 fused protein, the antibody titer of the serum sampled from the fish 3 weeks post-immunization were analyzed by ELISA. The results showed that the antibody titers for both the H1-D4ORF group and the D4ORF-VP3 group reached 1:6400–1:12,800 (Appendix A), but no specific antibody could be detected in the unvaccinated fish.

Moreover, fish 31 days post-initial vaccination were challenged by injection with CyHV-2. Five days later, PCR was used to assess whether the diseased fish were infected with CyHV-2. The infection rate was 85% in the control group injected with complete Freund’s adjuvant, and was 30% and 35% in the immunized groups vaccinated with H1-D4ORF and D4ORF-VP3, respectively. The RPS for immunized groups was 64.71% and 58.82% for H1-D4ORF and D4ORF-VP3, respectively (Table 1).

### 3.5. Baculovirus BmNPV Can Be Incorporated into BmCPV Polyhedra

As mentioned above, the polyhedra incorporating either H1-D4ORF or D4ORF-VP3 recombinant proteins can be used as a potential vaccine. Therefore, considering the safety of the vaccine, it was further determined whether recombinant baculovirus BmNPV can be embedded into BmCPV polyhedra when the BmCPV polh gene is co-expressed with H1-D4ORF or D4ORF-VP3 fused genes by the recombinant baculovirus BmNPV. The purified polyhedra from silkworm larvae infected with BmNPV-H1-D4ORF-polh or BmNPV-D4ORF-VP3-polh were lysed with 0.2 mol/L Na_2_CO_3_-NaHCO_3_ buffer (0.5 mL) to obtain a lysate of the polyhedra. The BmN cells were challenged by the lysate treated with 1 × PBS. At 4 days post-challenge, the cells showed cellular pathological changes, with cells becoming round and the polyhedra could be observed in challenged cells (Figure 5A,C). This morphological change might be due to the recombinant BmNPV infection. In contrast, the cells did not show any cytopathological effects when challenged with the lysate treated with 0.1% SDS (Figure 5B,D). Thus, the BmNPV envelope can be destroyed by treatment with 0.1% SDS (anionic detergents).

## 4. Discussion

Currently, PLGA, S-Alg, and chitosan are widely used as carriers for preparations of microencapsulated drugs and proteins [11]. Recently, cell protein crystallization has been used for structural biology and nanotechnology research [29]. In this study, BmCPVpolyhedra were used to encapsulate the recombinant vaccine against CyHV-2. The results presented here showed that the recombinant vaccine protein with either the H1 or VP3tag, co-expressed with BmCPVpolh in both the cultured BmN cells and silkworm larvae, could be incorporated into BmCPVpolyhedra. This result indicated that the microencapsulated vaccine can be generated by incorporating the target antigen into polyhedra. 

C. auratus gibelio gill hemorrhagic disease, caused by CyHV-2 infection, has spread widely in gibel carp aquaculture areas, with a 90–100% mortality rate [25,26,31]. Vaccine administration is believed to be the most effective approach to prevent and control C. auratus gibelio gill hemorrhagic disease. The RPS of gibel carp immunized with baculoviruses displaying and expressing ORF25, ORF25C, and ORF146 by the immersion route reached 83.3%, 87.5%, and 70.8%, respectively [30]. Recently, Huo et al. constructed recombinant plasmid pcORF25 with the ORF25 expression cassette and pcCCL35.2 containing a Ctenopharyngodonidella chemokine (C-C motif) ligand 35.2 (CCL35.2) expression cassette as a DNA vaccine and molecular adjuvant against CyHV-2. This study reports that the combination of CCL35.2 and pcORF25 significantly enhances the immune protection of gibel carp against CyHV-2 infection, with the highest survival rate being 70% [36]. In our previous studies, we constructed a recombinant baculovirus vector vaccine, BacCarassius-D4ORF, containing a fused codon-optimized sequence of D4ORF controlled by a β-actin promoter. The RPS of gibel carp immunized with BacCarassius-D4ORF via the oral or injection route reached 59.3% and 80.0%, respectively, following challenge [31]. In this study, it was found that specific antibodies against CyHV-2 could be detected 3 weeks post-immunization. The RPS of gibel carp immunized with the multi-subunit vaccine encapsulated by polyhedra via a single injection immunization reached 58.8–64.7% following challenge with CyHV-2. These results suggest that H1-D4ORF and D4ORF-VP3 recombinant proteins incorporated into polyhedra can be released in the fish, and thus C. auratus gibelio gill hemorrhagic disease can be prevented by vaccination with polyhedra incorporating antigen protein. However, the RPS of immunized fish with polyhedra incorporating H1-D4ORF and D4ORF-VP3 proteins was lower compared with previous reports. The different efficacy of these vaccines may be related to the slow release of antigens em-bedded in the polyhedra under physiological conditions [20,22], different types of vac-cines, different routes of administration and different detection time.

In this study, the BmCPVpolh gene was co-expressed with H1-D4ORF or D4ORF-VP3 fused genes by recombinant baculovirus BmNPV. In this case, there was a high level of cytoplasmic polyhedra in the blood of infected silkworms with the recombinant BmNPV. The yield in this study was 500 mg of polyhedra (wet weight) obtained from 40 mL of hemolymph from 100 infected silkworms. Purified polyhedra can be easily prepared by differential centrifugation from the hemolymph of infected silkworms. 

Polyhedra are resistant to degradation in the stomach because BmCPV polyhedra are resistant to enzyme action and can tolerate acidic treatment [15]. Therefore, polyhedra have the potential to be used as an oral vaccine carrier. Other studies showed that baculovirus vector vaccines against grass carp reovirus (GCRV) and CyHV-2 orally administered to grass carp, as well as gibel carp, induced good immunoprotection [31,34]. In a similar study, an oral Bacillus carrier vaccine against GCRV protected 50–60% of grass carp from infection with GCRV [37]. Moreover, orally immunized carp and koi with a genetically engineered Lactobacillus plantarum surface displaying G protein of the spring viremia of carp virus (SVCV) combined with the ORF81 protein of koi herpesvirus showed effective protection rates of 71% and 53%, respectively [38]. However, when carp received paraformaldehyde-fixed whole insect Sf9 cells expressing SVCV-G protein by oral vaccination with alginate-encapsulated cells, no protection against SVCV was detected [39]. These results indicated that vaccine carriers and the effective release of vaccine proteins from their carriers are crucial for oral vaccines. Although polyhedra are very stable, encapsulated protein kinase C [20], fibroblast growth factor (FGF)-2 [17,21,22], FGF-7 [17,22], epidermal growth factor [17,22], leukemia inhibitory factor [23], bone morphogenetic protein-2 [24], vascular endothelial growth factor [19], endostatin [19], and the major capsid protein VP1 of norovirus [18] were reported to be released from polyhedra both in vitro and in vivo.

BmCPV polyhedra can serve as the basis for the development of robust and versatile nanoparticles for biotechnological applications [40]. In this study, the results indicated that the H1-D4ORF and D4ORF-VP3 recombinant vaccine proteins encapsulated into the polyhedra were resistant to degradation at room temperature, suggesting that the H1-D4ORF and D4ORF-VP3 microencapsulated by polyhedra can be stored at room temperature.

Both the H1tag and the VP3tag allowed the fused protein to be incorporated into polyhedra [18,19,22,24]. In the present study, both H1-D4ORF and D4ORF-VP3 were detected in the supernatant of hemolymph from infected silkworm larvae with BmNPV-H1-D4ORF-polh and BmNPV-D4ORF-VP3-polh, respectively. This result indicated that some recombinant proteins were free, and located outside of the polyhedra. Some of the expressed Polh in the infected cells was crystallized to form polyhedra, while another portion was soluble, recombinant protein with the H1 tag or the VP3 tag that could be incorporated into polyhedra by the interaction of Polh with the H1 tag or the VP3 tag. Polyhedra are susceptive to alkali, and dissolve readily when exposed to pH values higher than 10.5 [15]. The recombinant protein with the H1tag or VP3tag could be specifically incorporated into cytoplasmic polyhedra. Therefore, when polyhedra are exposed to pH values higher than 10.5, the incorporated recombinant protein can be released. In this way, the recombinant protein with higher purity is relatively easy to obtain by adjusting the pH value to the isoelectric point (7.12) of the Polh in order to remove the Polh.

It was considered that the BmCPV virion can be specifically embedded intopolyhedra. However, it was found here that recombinant baculovirusBmNPV can be embedded into BmCPVpolyhedra. This result was similar to previous studies showing that the generated BmCPVpolyhedra in transformed cells can embed baculovirus BmPAK6 [33]. It is suggested that the polyhedra incorporating antigens are safe as a vaccine, because baculovirus with replication defect properties [41,42], as well as baculovirus, can be eliminated from the vaccinated fish [31].

## 5. Conclusions

The CyHV-2 subunit vaccine encapsulated by polyhedrin microcrystalline was successfully developed. And BmCPVpolyhedra can serve as the basis for the development of robust and versa-tile nanoparticles for biotechnological applications. The H1-D4ORF and D4ORF-VP3 microencapsulated by polyhedra can be stored at room temperature.

## Figures and Tables

**Figure 1 vaccines-09-00397-f001:**
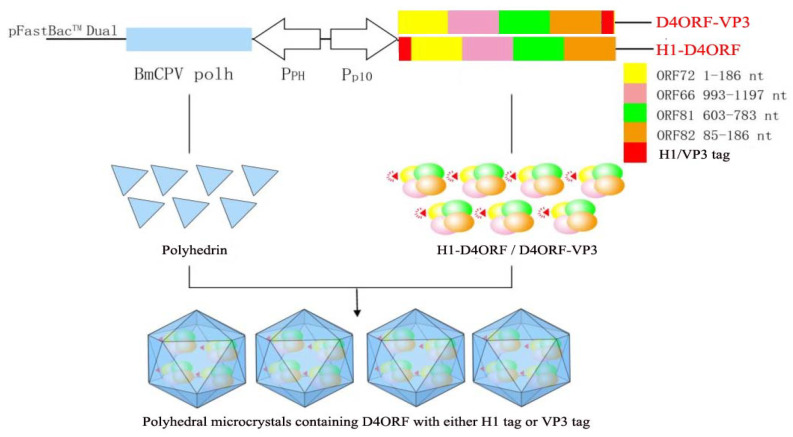
Strategies for constructing polyhedral microcrystals encapsulating antigen protein of CyHV-2. A codon-optimized sequence, D4ORF, comprising the ORF72 (region 1–186 nt), ORF66 (region 993–1197 nt), ORF81 (region 603–783 nt), and ORF82 (region 85–186 nt) genes of CyHV-2 was synthesized. The H1-D4ORF and D4ORF-VP3 sequences were, respectively, obtained by fusing the H1-helix sequence (region 1–90 nt) of the BmCPVpolh to the 5′ terminal end of D4ORF and by fusing a partial sequence (1–279 nt) of the BmCPV VP3 gene to the 3′ terminal end of D4ORF, respectively. The H1-D4ORF or D4ORF-VP3 sequences were driven by the baculovirus p10 promoter (Pp10). The BmCPV polh was controlled by the baculoviruspolh promoter (PPH). When either the H1-D4ORF or D4ORF-VP3 wasco-expressed with BmCPVpolh in the cell, the respective H1-D4ORF and D4ORF-VP3 proteins were incorporated into polyhedral microcrystals.

**Figure 2 vaccines-09-00397-f002:**
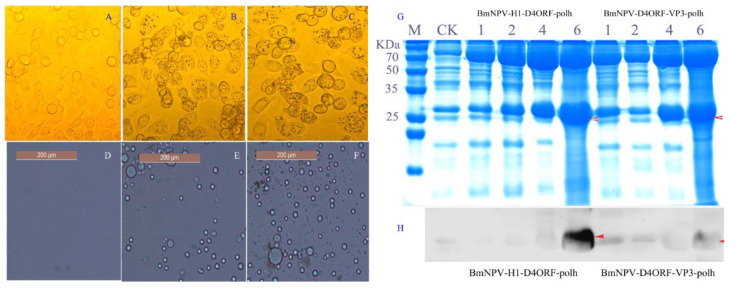
H1-D4ORF, D4ORF-VP3, and BmCPVpolh expressed in the BmN cells and silkworm larvae. (**A**) The uninfected BmN cells; (**B**) BmCPV polyhedra were observed in cells transfected with Bacmid-H1-D4ORF-polh; (**C**) BmCPV polyhedra were observed in cells transfected with Bacmid-D4ORF-VP3-polh; (**D**) BmCPV polyhedra were not found in the hemolymph of uninfected silkworm larvae; (**E**) BmCPV polyhedra generated in the hemolymph of infected silkworm larvae with BmNPV-H1-D4ORF-polh; (**F**) BmCPV polyhedra generated in the hemolymph of infected silkworm larvae with BmNPV-D4ORF-VP3-polh; (**G**) SDS-PAGE of the hemolymph from silkworm larvae infected with BmNPV-H1-D4ORF-polh/BmNPV-D4ORF-VP3-polh. Lane M, protein marker; lane CK, the hemolymph of uninfected silkworm larvae; lane 1, 2, 4, and 6, the hemolymph of silkworm larvae infected with BmNPV-H1-D4ORF-polh/BmNPV-D4ORF-VP3-polh 1, 2, 4, and 6 days post-infection. The recombinant proteins areindicated by an arrowhead; (**H**) Western blot analysis depicting the hemolymph of silkworm larvae infected with BmNPV-H1-D4ORF-polh/BmNPV-D4ORF-VP3-polh. Mouse anti-ORF72 (1:500) was used as the primary antibody and an HRP-conjugated goat anti mouse IgG (1:20,000) was used as the second antibody. The signal bands representing recombinant proteins are indicated by an arrowhead.

**Figure 3 vaccines-09-00397-f003:**
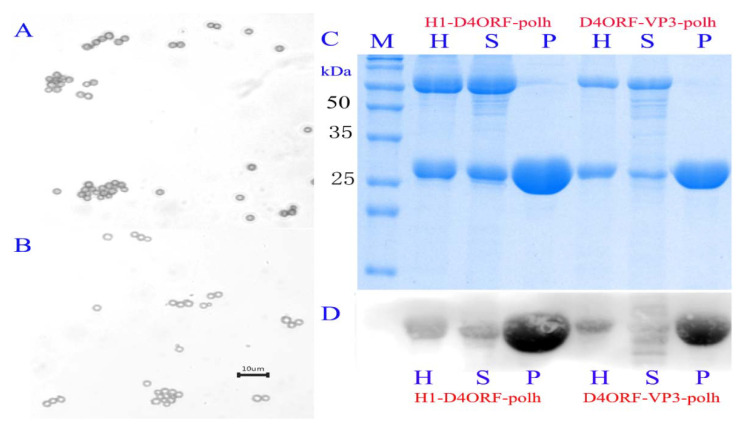
H1-D4ORF and D4ORF-VP3 recombinant proteins can be incorporated into BmCPV polyhedra. (**A**,**B**) are the purified recombinant cytoplasmic polyhedra generated in BmNPV-H1-D4ORF-polh- and BmNPV-D4ORF-VP3-polh-infected silkworm larvae, respectively; (**C**) SDS-PAGE of hemolymph, supernatant of hemolymph, and purified recombinant cytoplasmic polyhedra. The samples were from silkworm larvae 6 days post-infection with BmNPV-H1-D4ORF-polh/BmNPV-D4ORF-VP3-polh. Lane M, protein marker; lane H, hemolymph; S, supernatant of hemolymph; P, purified recombinant cytoplasmic polyhedra; (**D**) Western blot analysis of the hemolymph, supernatant of the hemolymph, and purified recombinant cytoplasmic polyhedra. The samples were the same as those of SDS-PAGE. Mouse anti-ORF72 (1:500) was used as primary antibody and an HRP-conjugated goat anti mouse IgG (1:20,000) was used as the secondary antibody.

**Figure 4 vaccines-09-00397-f004:**
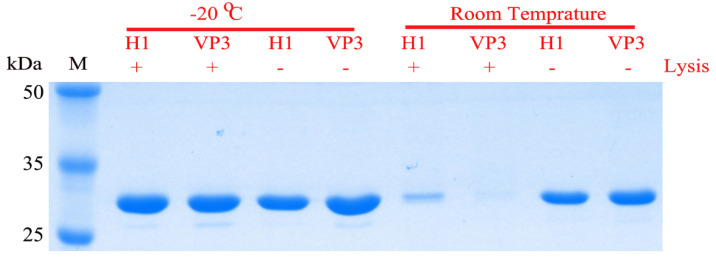
SDS-PAGE for the preserved recombinant polyhedra and their lysis under different conditions. Lane M, protein marker; +, lysed polyhedra; −, polyhedra without lysis; H1, BmNPV-H1-D4ORF-polh polyhedra, VP3, BmNPV-D4ORF-VP3-polh polyhedra. The polyhedra and their lysates were stored at −20 °C or room temperature (about 22 °C) for 2 weeks.

**Figure 5 vaccines-09-00397-f005:**
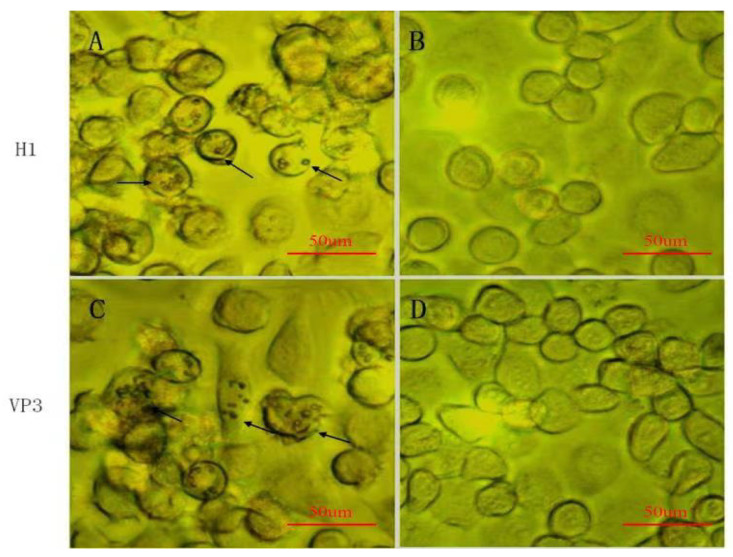
Detection of baculovirus incorporated into BmCPV polyhedra. (**A**) The inoculated BmN cells with BmNPV-H1-D4ORF-polh polyhedra lysis without 0.1% SDS treatment; (**B**) the inoculated BmN cells with BmNPV-H1-D4ORF-polh polyhedra lysis with 0.1% SDS treatment; (**C**) the inoculated BmN cells with BmNPV-D4ORF-VP3-polh polyhedra lysis without 0.1% SDS treatment; (**D**) the inoculated BmN cells with BmNPV-D4ORF-VP3-polh polyhedralysis with 0.1% SDS treatment. H1, BmNPV-H1-D4ORF-polh polyhedra; VP3, BmNPV-D4ORF-VP3-polh polyhedra. The cytoplasmic polyhedra are indicated by an arrowhead.

**Table 1 vaccines-09-00397-t001:** Immunoprotection of CyHV-2 infection by immunological injection of recombinant polyhedra.

Group	Control	H1-D4ORF	D4ORF-VP3
Injection Dose	PBS	7 × 106 per fish	4.4 × 106 per fish
Infection Rate (%)	17/20	6/20	7/20
Relative Percent Survival (%)	/	64.7	58.82

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
