# Peer review of "Incidence of Carassius auratus Gibelio Gill Hemorrhagic Disease Caused by CyHV-2 Infection Can Be Reduced by Vaccination with Polyhedra Incorporating Antigens"

_vaccines, 2021, doi:10.3390/vaccines9040397_

Round 1

Reviewer 1 Report

Manuscript title: Incidence of Carassius auratus gibelio gill hemorrhagic disease 2 caused by CyHV-2 infection can be reduced by vaccination 3 with polyhedra incorporating antigens

The manuscript described development of a vaccine using the baculovirus system. The author describes the development of a recombinant virus expressing different ORFs of CyHV-2 which was subsequently used for immunized fish against the virus. An antibody ELISA was employed to show the development of the specific antibodies. The authors also tested the stability of the polyhedra and suggested the use the polyhedra as antigen microencapsulator and a novel formulation strategy.

The study is interesting but, in my opinion, there are some issues that need fixing before the manuscript is considered suitable for publication.

Main comments:

  • The presentation of the data is not appropriate. The figure texts should be self-explanatory (i.e. the figures could be understood by only reading the figure text). This is not the case and in some cases the figure text is imbedded in the main text (e.g. line 278-292).
  • The SDS-PAGE and western blot results in figure 2 (C and D) are lacking the negative control.
  • In line 328 it is stated that the protein concentration/stability after incubation at room temperature is similar to -20°C in the non-lysed samples. In my opinion, given that equal amounts of proteins are aliquoted and analyzed after storage, the result presented in Fig.4 actually indicate that samples stored at -20°C had much higher protein than the RT ones. A densitometric analysis is necessary to quantitively show the differences.
  • Results section 3.5: this section is not clearly written, and I failed to understand how the samples were treated to remove the observed toxic effects. When and how the 0.1 SDS was performed is neither described in the M&M nor in this section. In the M&M it is only described in the procedure for purifying the polyhedral. Consequently, the claim that the virus is inactivated is by 0.1% SDS is not supported by the data in the current form.
  • Why only 3 samples were analyzed with ELISA. Serum from at least 6 fish should be analyzed to get a biologically relevant data.
  • The discussion is somewhat lengthy in unnecessary way in some of the paragraphs (e.g. paragraph 2 and 4) while some important points have not been discussed in my opinion. I suggest adding a discussion about:
  • The use and the testing of polyhedra as a microincapsulator.
  • The observed toxicity of the polyhedra in vitro and in vivo.
  • The safety of using the recombinant baculoviruses and subsequently the suggested vaccination strategy.
  • Inactivation of the recombinant virus.
  •  

Minor comments:

Line 34 : microencapsulated instead of microcapsuled.

Line 52: change to, due to exposure to various enzymes….

Line 148: please describe the blocking step and add the primary antibody dilution

Line 197: change the blank control to untreated control

Lines 229-231: please remove the irrelevant text

Author Response

Dear editors and reviewers,

On behalf of my co-authors, we thank you very much for giving us an opportunity to revise our manuscript, we appreciate editor very much for positive and constructive comments and suggestions on our manuscript entitled “Incidence of Carassius auratus gibelio gill hemorrhagic disease caused by CyHV-2 infection can be reduced by vaccination with polyhedra incorporating antigens”. We have made revision which marked Red in the paper. We have tried our best to revise our manuscript according to the comments. Attached please find the revised version, which we would like to submit for your kind consideration. We would like to express our great appreciation to you for comments on our paper. Looking forward to hearing from you. Thank you and best regards.

Best wishes,

Corresponding author: Chengliang Gong

E-mail address: [email protected]

Reviewer 1:

Main comments:

  1. The presentation of the data is not appropriate. The figure texts should be self-explanatory (i.e. the figures could be understood by only reading the figure text). This is not the case and in some cases the figure text is imbedded in the main text (e.g. line 278-292).

Response: Thanks for your comments. We have revised the legends of Figures in the main text.

  1. The SDS-PAGE and western blot results in figure 2 (C and D) are lacking the negative control.

Response: Thanks for your comments. It is a pity that the negative control was lost in the SDS-PAGE and western blot results in figure 2. While, the hemolymph of uninfected silkworm larvae was applied as the CK. From the SDS-PAGE results, we could clearly observe the increasing of the expression level of D4ORF following the viral infection process. From the SDS-PAGE and western blotting results, we could conclude that the H1-D4ORF (27.94 kDa) and D4ORF-VP3 (34.87 kDa) recombinant proteins could be respectively detected in the hemolymph of BmNPV-H1-D4ORF-polh and BmNPV-D4ORF-VP3-polh infected silkworms.

  1. In line 328 it is stated that the protein concentration/stability after incubation at room temperature is similar to -20°C in the non-lysed samples. In my opinion, given that equal amounts of proteins are aliquoted and analyzed after storage, the result presented in Fig.4 actually indicate that samples stored at -20°C had much higher protein than the RT ones. A densitometric analysis is necessary to quantitively show the differences.

Response: Thanks for your comments. We have provided the densitometric analysis of the Figure 4.

  1. Results section 3.5:this section is not clearly written, and I failed to understand how the samples were treated to remove the observed toxic effects. When and how the 0.1 SDS was performed is neither described in the M&M nor in this section. In the M&M it is only described in the procedure for purifying the polyhedral. Consequently, the claim that the virus is inactivated is by 0.1% SDS is not supported by the data in the current form.

Response: Thanks for your comments. The procedure for purifying the polyhedral was described in our previous published papers (Integrin beta and receptor for activated protein kinase C are involved in the cell entry of Bombyx mori cypovirus), here we provided the reference in the section of M&M. We have revised the conlcudion of results section 3.5: BmNPV envelope can be destroyed by treatment with 0.1% SDS (anionic detergents).

  1. Why only 3 samples were analyzed with ELISA. Serum from at least 6 fish should be analyzed to get a biologically relevant data.

Response: Thanks for your comments. For biologically relevant data, more samples are better. In our future work, we will use more samples for ELISA analysis.

  1. The discussion is somewhat lengthy in unnecessary way in some of the paragraphs (e.g. paragraph 2 and 4) while some important points have not been discussed in my opinion. I suggest adding a discussion about:

Response: Thanks for your comments. We have revised the corresponding points in the section of discussion.

  1. The use and the testing of polyhedra as a microincapsulator. The observed toxicity of the polyhedra in vitro and in vivo. The safety of using the recombinant baculoviruses and subsequently the suggested vaccination strategy. Inactivation of the recombinant virus.

Response: Thanks for your comments. We have revised the corresponding points in the section of discussion.

  1. Line 34 : microencapsulated instead of microcapsuled.

Response: Thanks for your comments. We have revised them in the text.

  1. Line 52: change to, due to exposure to various enzymes….

Response: Thanks for your comments. We have revised it the text.

  1. Line 148: please describe the blocking step and add the primary antibody dilution

Response: Thanks for your comments. We have revised them in the section of M&M.

  1. Line 197: change the blank control to untreated control

Response: Thanks for your comments. We have revised it.

  1. Lines 229-231: please remove the irrelevant text

Response: Thanks for your comments. We have removed the irrelevant text.

Reviewer 2 Report

The present manuscript titled “Incidence of Carassius auratus gibelio gill hemorrhagic disease caused by CyHV-2 infection can be reduced by vaccination with polyhedral incorporating antigens” is a really interesting approach for the future use of a new vaccine delivery system. The investigation establishes the basis for further studies in which antigens could be encapsulated within protein microcrystals (polyhedral), designed recombinant proteins. The authors showed how to construct the recombinant vaccine, demonstrated its stability and the conferred protection in fish after a viral challenge. The topic is really worthy to investigate. The manuscript is very-well written, all the sections are developed in a very clear way. I have only some minor comments that could improve the manuscript. Once the revision is done, the manuscript is suitable for publication.

Abstract

Line 33: Typo, “that” is written twice

Introduction

Line 79: The authors should include the reference for the previous studies. I guess the reference is the number 16, but they should include it.

Line 84: Typo, dot at the end of the sentence (after (17)).

Materials and methods

Line 134: Escherichia coli in italic

Line 141: For sure I missed something, but the authors should explain with more detail which are the BmN cells.

Line 151: B. mori should not be in italic, since it is included in a sentence in italic

Line 155: The authors should include the employed antibodies for the Western-blot.

Lines 165-167: These lines could be deleted and refer to the section 2.5.

Lines 169, 223: Typo, subscripts for the chemical formula  

Lines 191, 194, 211, 223, 226: Typo, superscripts for the 10^6, 10^7, 10^8, 10^5

Lines 228-231: Typo, the authors by mistake included the guide for writing the section of Results.

Results

Line 242: The authors referred to Figure 1-A, but there is only Figure 1 (without sections)

Line 246: Typo, Escherichia coli in italic

Lines 268-292: I am not sure why the figure legends for Figures 1 and 2 are written here and not just below the figures.

Lines 309-318: I am not sure why the figure legends for Figure 3 re written here and not just below the figure.

Lines 321-324: If the authors are describing the differences between the samples at -20 ºC, regardlees the samples were lysed or not, the groups LPS-20 and LPS-RT are not the groups to compare, since both groups are consisting in lysed samples.

Lines 324-326: If the authors want to compare samples stored at room temperature, the groups are not PS-20 and PS-RT.

Figure 4: Typo, “Temperature”

Line 357: Typo, subscripts for the chemical formula 

Discussion

Line 448: Italic “Bacillus

Line 499: Typo “baculoviruses”

Author Response

Dear editors and reviewers,

On behalf of my co-authors, we thank you very much for giving us an opportunity to revise our manuscript, we appreciate editor very much for positive and constructive comments and suggestions on our manuscript entitled “Incidence of Carassius auratus gibelio gill hemorrhagic disease caused by CyHV-2 infection can be reduced by vaccination with polyhedra incorporating antigens”. We have made revision which marked Red in the paper. We have tried our best to revise our manuscript according to the comments. Attached please find the revised version, which we would like to submit for your kind consideration. We would like to express our great appreciation to you for comments on our paper. Looking forward to hearing from you. Thank you and best regards.

Best wishes,

Corresponding author: Chengliang Gong

E-mail address: [email protected]

Reviewer 2:

  1. Line 33: Typo, “that” is written twice.

Response: Thanks for your comments. We have revised it in the text.

  1. Line 79: The authors should include the reference for the previous studies. I guess the reference is the number 16, but they should include it. Line 84: Typo, dot at the end of the sentence (after (17)).

Response: Thanks for your comments. We have revised them in the text.

  1. Line 134: Escherichia coli in italic Line 141: For sure I missed something, but the authors should explain with more detail which are the BmN cells.Line 151:  morishould not be in italic, since it is included in a sentence in italic Line 155: The authors should include the employed antibodies for the Western-blot. Lines 165-167: These lines could be deleted and refer to the section 2.5. Lines 169, 223: Typo, subscripts for the chemical formula  Lines 191, 194, 211, 223, 226: Typo, superscripts for the 10^6, 10^7, 10^8, 10^5 Lines 228-231: Typo, the authors by mistake included the guide for writing the section of Results.

Response: Thanks for your comments. We have revised them in the Materials and methods.

  1. Line 242: The authors referred to Figure 1-A, but there is only Figure 1 (without sections)

Response: Thanks for your comments. We have revised it in the text.

  1. Line 246: Typo, Escherichia coli in italic

Response: Thanks for your comments. We have revised it in the text.

  1. Lines 268-292: I am not sure why the figure legends for Figures 1 and 2 are written here and not just below the figures.

Response: Thanks for your comments. We have revised them in the text.

  1. Lines 309-318: I am not sure why the figure legends for Figure 3 re written here and not just below the figure.

Response: Thanks for your comments. We have revised it in the text.

  1. Lines 321-324: If the authors are describing the differences between the samples at -20 ºC, regardlees the samples were lysed or not, the groups LPS-20 and LPS-RT are not the groups to compare, since both groups are consisting in lysed samples. Lines 324-326: If the authors want to compare samples stored at room temperature, the groups are not PS-20 and PS-RT.

Response: Thanks for your comments. To assess that the polyhedral microcrystals can protect the recombinant protein and avoid them degradation, figure 4 only showed that regardless of whether the polyhedra were lysed or not, there was no significant change in the electrophoretic pattern between group LPS-20 and group LPS-RT.

  1. Figure 4: Typo, “Temperature”

Response: Thanks for your comments. We have revised it in the figure 4.

  1. Line 357: Typo, subscripts for the chemical formula 

Response: Thanks for your comments. We have revised it. 

  1. Line 448: Italic “Bacillus” Line 499: Typo “baculoviruses”

Response: Thanks for your comments. We have revised them in the section of Discussion. 

Reviewer 3 Report

The manuscript by Tingting Zhanga and colleagues presents and original work about the use of an anti-CyHV-2 multi-subunit vaccine protein encapsulated into BmCPV polyhedra to protect gibel carp from CyHV-2. They found that the encapsulated vaccine protein into polyhedra can be protected from degradation and the relative percent survival (RPS) of gibel carp immunized with the polyhedra via injection route reached 64.71% and 58.82%, following challenge with CyHV-2. These results suggest that incorporating vaccine protein into BmCPV polyhedra may be a novel approach for developing aquaculture microcapsuled vaccines.

I found this article really interesting and well written. However, it was not properly structured, figures and legends are disordered. For example, figure 1 appears in page 6 and its legend appear in the middle of the manuscript from line 268 to 277. Please check every figure and legend. It makes so difficult to understand the manuscript.

-Figure 2 D, E and F,  I can not see what exactly I should see. Please explain better.

-Figure 2 H I think authors should include in the western blot a housekeeping protein in order to normalized the quantity of protein.

Figure 3-Legend A and B explain better. Is this a microscopy image? Please clarified. Indicate the magnification of the images.

Figure5 legend, should indicate the magnification of the image and indicate that this is a microscopy image.

In the supplementary material, figure S2, indicate the statistical analysis in the legend.

Apart from that, I have some questions and minor comments for the authors.

  • Question:

1-In the manuscript gibel carp immunized with the polyhedra via injection route reached 64.71% and 58.82% of protection, following challenge with CyHV-2. Have you check the innate immune response in gibel carp immunized with the polyhedral at early times?.  Have the polyedra alone some immune response in injected fish?.

2-If the polyedra vaccine is incorporated into diet by pellet, Do you know if they have some effect in the gut microbiota of fish?

  • Minor comments. I have found many mistakes please change them.
  • Line 116 authors should indicate the commercial reference of the filters used.
  • Line 119 authors said 105 copies/μL, maybe this is 105 copies/μL?.
  • Define the acronym ORF and nt the first time they appear in the manuscript.
  • Line 141 BmN cells. Authors should define which kind of cells they are?.
  • Line 141 authors said 105 cells in 200 μ Is this correct or maybe 105?.
  • Line 141 TC-100 medium. Authors should indicate the commercial reference.
  • Line 146 authors said 50 μg of protein from infected cells. I am confused but I was thinking that cells were transfected with Fugene not infected. Please correct it.
  • Line 147 polyvinylidene difluoride (PVDF) membranes. Authors should indicate the commercial reference.
  • Line 161 SDS. Authors should indicate the commercial reference.
  • In material and methods the point 2.7 authors do not explain how the polyhedra is lysed. Please add it.
  • Line 182 Indicate commercial reference of Coomasie blue.
  • Line 189 define group CK.
  • Line 191 and 194 0 × 106 polyhedra  and 4.4 × 106 polyhedra maybe is7.0 and4.4  × 106?. Please correct them.
  • Line 206 Indicate commercial reference of Tween-20.
  • Line 211 CyHV-2 stock (107 copies of virus per fish). Maybe, 107?. Please correct it.
  • Line 223purified polyhedra (108 in 0.5 mL) or 108?.
  • Line 226 2 × 105 or 105.
  • Line 228 changes of the cells by microscopy.3. Results. I do not understand the sentence. Also include commercial reference of the microscope.
  • From line 229 to 231 “This section may be divided by subheadings. It should provide a concise and precise description of the experimental results, their interpretation, as well as the experimental conclusions that can be drawn”.  This sentence is a mistake so it should be removed from the manuscript.
  • Line 242 Figure 1 A does not exit, only figure 1. Please remove A.
  • Line 404 “vaccine. And”. Please remove the dot.

Author Response

Dear editors and reviewers,

On behalf of my co-authors, we thank you very much for giving us an opportunity to revise our manuscript, we appreciate editor very much for positive and constructive comments and suggestions on our manuscript entitled “Incidence of Carassius auratus gibelio gill hemorrhagic disease caused by CyHV-2 infection can be reduced by vaccination with polyhedra incorporating antigens”. We have made revision which marked Red in the paper. We have tried our best to revise our manuscript according to the comments. Attached please find the revised version, which we would like to submit for your kind consideration. We would like to express our great appreciation to you for comments on our paper. Looking forward to hearing from you. Thank you and best regards.

Best wishes,

Corresponding author: Chengliang Gong

E-mail address: [email protected]

Reviewer 3:

  1. figure 1 appears in page 6 and its legend appear in the middle of the manuscript from line 268 to 277. Please check every figure and legend. It makes so difficult to understand the manuscript.

Response: Thanks for your comments. We have revised them in the text. 

  1. Figure 2 D, E and F,  Ican not see what exactly I should see. Please explain better.

Response: Thanks for your comments. We have revised them in the legend of Figure 2.

  1. Figure 2 H I think authors should include in the western blot a housekeeping protein in order to normalized the quantity of protein.

Response: Thanks for your comments. It’s a pity that a housekeeping protein was missed in the western blot.

  1. Figure 3-Legend A and B explain better. Is this a microscopy image? Please clarified. Indicate the magnification of the images.

Response: Thanks for your comments. We have explained in the legends.

  1. Figure5 legend, should indicate the magnification of the image and indicate that this is a microscopy image.

Response: Thanks for your comments. We have explained in the legends.

  1. In the manuscript gibel carp immunized with the polyhedra via injection route reached 64.71% and 58.82% of protection, following challenge with CyHV-2. Have you check the innate immune response in gibel carp immunized with the polyhedral at early times?.  Have the polyedra alone some immune response in injected fish?.

Response: Thanks for your comments. These works was not provided in this manuscript.

  1. If the polyedra vaccine is incorporated into diet by pellet, Do you know if they have some effect in the gut microbiota of fish?

Response: Thanks for your comments. We will have a try.

  1. Line 116 authors should indicate the commercial reference of the filters used. Line 141 TC-100 medium. Authors should indicate the commercial reference. Line 147 polyvinylidene difluoride (PVDF) membranes. Authors should indicate the commercial reference. Line 161 SDS. Authors should indicate the commercial reference. Line 182 Indicate commercial reference of Coomasie blue. Line 206 Indicate commercial reference of Tween-20.

Response: Thanks for your comments. We have provided the commercial references.

  1. Define the acronym ORF and nt the first time they appear in the manuscript.

Response: Thanks for your comments. We have revised them.

  1. Line 141 BmN cells. Authors should define which kind of cells they are?.

Response: Thanks for your comments. Cultured silkworm ovary-derived BmN cell line.

  1. Line 146 authors said 50 μg of protein from infected cells. I am confused but I was thinking that cells were transfected with Fugene not infected. Please correct it.

Response: Thanks for your comments. We used the recombinant baculoviruses BmNPV-H1-D4ORF-polh or BmNPV-D4ORF-VP3-polh to infect the BmN cells or not the plasmids.

  1. In material and methods the point 2.7 authors do not explain how the polyhedra is lysed. Please add it.

Response: Thanks for your comments. We have provided our published method (Integrin beta and receptor for activated protein kinase C are involved in the cell entry of Bombyx mori cypovirus).

  1. Line 189 define group CK.

Response: Thanks for your comments. We have revised it.

  1. Line 119 authors said 105 copies/μL, maybe this is 105copies/μL?. Line 141 authors said 105 cells in 200 μ Is this correct or maybe 105? Line 191 and 194 0 × 106 polyhedra  and 4.4 × 106 polyhedra maybe is7.0 and4.4  × 106?. Please correct them. Line 211 CyHV-2 stock (107 copies of virus per fish). Maybe, 107?. Please correct it. Line 223purified polyhedra (108 in 0.5 mL) or 108?. Line 226 2 × 105 or 105.

 Response: Thanks for your comments. We have corrected them.

  1. Line 228 changes of the cells by microscopy.3. Results. I do not understand the Also include commercial reference of the microscope.

Response: Thanks for your comments. The commercial reference of the microscope was provided.

  1. From line 229 to 231 “This section may be divided by subheadings. It should provide a concise and precise description of the experimental results, their interpretation, as well as the experimental conclusions that can be drawn”.  This sentence is a mistake so it should be removed from the manuscript.

Response: Thanks for your comments. We have removed this sentence from the manuscript.

  1. Line 242 Figure 1 A does not exit, only figure 1. Please remove A.

Response: Thanks for your comments. We have revised it.

  1. Line 404 “vaccine. And”. Please remove the dot.

Response: Thanks for your comments. We have revised it.

Round 2

Reviewer 1 Report

I appreciate the author's effort but They haven't responded to most of the important comments except for the ones concerning the figure texts.

  • The authors did not provide a negative control in figure 3. This is necessary because a non specific band is seen in figure 1. The authors refer to their previous publication where these antibodies has been used. However, no proper characterization for these antibodies was done then either and it is not clear for me where these antibodies are commercially produced or custom-made.
  • The densitometry analysis was not performed in figure 4 although the authors indicate so in their response letter.
  • The issue raised previously about result section 3.5 has not been resolved or answered. the authors just refer to their previous publication where the issue of SDS treatment has not been specifically addressed. here is the comment I gave earlier: Results section 3.5:this section is not clearly written, and I failed to understand how the samples were treated to remove the observed toxic effects. When and how the 0.1 SDS was performed is neither described in the M&M nor in this section. In the M&M it is only described in the procedure for purifying the polyhedral. Consequently, the claim that the virus is inactivated is by 0.1% SDS is not supported by the data in the current form.
  • The biological replicates in ELISA have not been increased. This is important in order to get biologically relevant data in fish experiments where the standard deviation is often high. Samples should be stored and available for some years even after publications and the ELISA run does not take more than 1-2 days.
  • The discussion has not been significantly improved.

Author Response

Dear editors and reviewers,

On behalf of my co-authors, we thank you very much for giving us an opportunity to revise our manuscript, we appreciate editor very much for positive and constructive comments and suggestions on our manuscript entitled “Incidence of Carassius auratus gibelio gill hemorrhagic disease caused by CyHV-2 infection can be reduced by vaccination with polyhedra incorporating antigens”. We have made revision which marked yellow in the paper. We have tried our best to revise our manuscript according to the comments. Attached please find the revised version, which we would like to submit for your kind consideration. We would like to express our great appreciation to you for comments on our paper. Looking forward to hearing from you. Thank you and best regards.

Best wishes,

Corresponding author: Chengliang Gong

E-mail address: [email protected]

  •  
  • The authors did not provide a negative control in figure 3. This is necessary because a non specific band is seen in figure 1. The authors refer to their previous publication where these antibodies has been used. However, no proper characterization for these antibodies was done then either and it is not clear for me where these antibodies are commercially produced or custom-made.

Response: Thanks for your comments. We think this negative control is not necessary. In the Figure 1 is a diagram of how to contruct the polyhedral microcrystals encapsulated antigen protein of CyHV-2. We don’t use antibody in this part. The specificity of antibodies used in this study were same with our previous used in published paper (Recombinant baculovirus BacCarassius-D4ORFs has potential as a live vector vaccine against CyHV-2). Anti-ORF72 was home-made by our lab as followed.

  • The densitometry analysis was not performed in figure 4 although the authors indicate so in their response letter.

Response: Thanks for your comments. We have not obtained a perfect result from the densitometry analysis. From the figure 4, we could conclude that the protein was degraded quickly in the sample without the protection from the polyhedra.

  • The issue raised previously about result section 3.5 has not been resolved or answered. the authors just refer to their previous publication where the issue of SDS treatment has not been specifically addressed. here is the comment I gave earlier: Results section 3.5:this section is not clearly written, and I failed to understand how the samples were treated to remove the observed toxic effects. When and how the 0.1 SDS was performed is neither described in the M&M nor in this section. In the M&M it is only described in the procedure for purifying the polyhedral. Consequently, the claim that the virus is inactivated is by 0.1% SDS is not supported by the data in the current form.

Response: Thanks for your comments. The procedure for purifying the polyhedral was described in our previous published papers (Integrin beta and receptor for activated protein kinase C are involved in the cell entry of Bombyx mori cypovirus), here we provided the reference in the section of M&M. We have revised the conclusion of results section 3.5: BmNPV envelope can be destroyed by treatment with 0.1% SDS (anionic detergents). In the whole procedure, we have not use any methods to remove the observed toxic effects. The methods used in this manuscript was a normal procedure in the subject of BmNPV or BmCPV. Line 223-226, we have described the treatment with 0.1% SDS. The purified polyhedra (108 in 0.5 mL) were lysed with 0.2 mol/L Na2CO3-NaHCO3 buffer (0.5 mL) at 30 °C for 30 min, and mixed with 0.1% SDS (1 mL) or PBS (1 mL), followed by incubation for 30 min at 26 °C.

  • The biological replicates in ELISA have not been increased. This is important in order to get biologically relevant data in fish experiments where the standard deviation is often high. Samples should be stored and available for some years even after publications and the ELISA run does not take more than 1-2 days.

Response: Thanks for your comments. We have described clearly about them in the M&M. In group LPS-20, the lysed polyhedra were stored at 20 °C; in group LPS-RT, the lysed polyhedra were stored at room temperature (about 22 °C); in group PS-20, polyhedra without lysis were stored at 20 °C; and in group PS-RT, polyhedra without lysis were stored at room temperature. After two weeks, 15 μg of the treated samples were subjected to SDS-PAGE. Protein degradation was estimated by electropherogram after the PAGE gel was stained with Coomassie brilliant blue R250 (Sangon, Shanghai, China).

  • The discussion has not been significantly improved.

Response: Thanks for your comments. We have tried out best to revise the section of discussion.

Round 3

Reviewer 1 Report

The authors did not respond to the comments. Of particular interest is the control in figure 3C because it shows that the purified protein was not present  in the hemolymph obtained normal worm. The authors did not give a proper explanation for not presenting the control and I found it difficult to explain. In addition, the antibody levels in serum obtained from only 3 fish were presented. I also find it difficult to explain why the authors did not analyse more serum samples. Both results should be easy to obtain.

The comments regarding result section 3.5 has not been considered either.

Author Response

Dear editors and reviewers,

On behalf of my co-authors, we thank you very much for giving us an opportunity to revise our manuscript, we appreciate editor very much for positive and constructive comments and suggestions on our manuscript entitled “Incidence of Carassius auratus gibelio gill hemorrhagic disease caused by CyHV-2 infection can be reduced by vaccination with polyhedra incorporating antigens”. We have made revision which marked Red in the paper. We have tried our best to revise our manuscript according to the comments. Attached please find the revised version, which we would like to submit for your kind consideration. We would like to express our great appreciation to you for comments on our paper. Looking forward to hearing from you. Thank you and best regards.

Best wishes,

Corresponding author: Chengliang Gong

E-mail address: [email protected]

Comments:

1.The authors did not respond to the comments. Of particular interest is the control in figure 3C because it shows that the purified protein was not present  in the hemolymph obtained normal worm. The authors did not give a proper explanation for not presenting the control and I found it difficult to explain. In addition, the antibody levels in serum obtained from only 3 fish were presented. I also find it difficult to explain why the authors did not analyse more serum samples. Both results should be easy to obtain.

Response:Thanks for your comments. First of all, your suggestion is very reasonable, we have confirmed that ORF72 is specific and non-existent in silkworm larvae in figure 2. On this basis, we further carried out the experiment of figure 3C, so we did not present the control here.

The antibody levels in serum obtained from only 3 fish were presented. We consider that the 3 fish randomly selected is representative and statistical significance. But your suggestion is also very correct and more samples can explain the problem.

2.The comments regarding result section 3.5 has not been considered either.

Response:Thanks for your comments. We have revised the conclusion of results section 3.5: BmNPV envelope can be destroyed by treatment with 0.1% SDS (anionic detergents). In the whole procedure, we have not use any methods to remove the observed toxic effects. The methods used in this manuscript was a normal procedure in the subject of BmNPV or BmCPV. Line 223-226, we have described the treatment with 0.1% SDS. The purified polyhedra (108 in 0.5 mL) were lysed with 0.2 mol/L Na2CO3-NaHCO3 buffer (0.5 mL) at 30 °C for 30 min, and mixed with 0.1% SDS (1 mL) or PBS (1 mL), followed by incubation for 30 min at 26 °C.